# The Developmental Pathways of Preschool Children with Acute Lymphoblastic Leukemia: Communicative and Social Sequelae One Year after Treatment

**DOI:** 10.3390/children6080092

**Published:** 2019-08-13

**Authors:** Marta Tremolada, Livia Taverna, Sabrina Bonichini, Marta Pillon, Alessandra Biffi

**Affiliations:** 1Department of Development and Social Psychology, University of Padua, 35131 Padova, Italy; 2Department of Women and Child’s Health, Pediatric Hematology, Oncology and Stem Cell Transplant Center, University of Padua, 35127 Padova, Italy; 3Faculty of Education, Free University of Bozen-Bolzano, 39042 Brixen-Bressanone, Italy

**Keywords:** preschool, leukemia, adaptive behavior, developmental skills, healthy peers

## Abstract

Early childhood is considered to be a period of rapid development, with the acquisition of abilities predicting future positive school competences. Motor, cognitive, and social difficulties related to cancer therapies heavily impact the development of children with cancer. This study focused on two main aims: To assess the developmental pathways of preschool children with acute lymphoblastic leukemia one year post-treatment and to compare these abilities both with those of a control group of healthy peers and with Italian norms. Forty-four children and their families, recruited through the Hematology-Oncologic Clinic of the Department of Child and Woman Health (University of Padua), agreed to participate in this study. The children’s mean age was 4.52 years (SD = 0.94, range = 2.5–6 years), equally distributed by gender, all diagnosed with acute lymphoblastic leukemia. Matched healthy peers were recruited through pediatricians’ ambulatories. Each family was interviewed adopting the Vineland adaptive behavior scales. Paired sample Wilcoxon tests revealed that children were reported to have significantly more developmental difficulties than their healthy peers. When compared with Italian norms, they scored particularly low in verbal competence, social, and coping skills. No significant association was found between treatment variables and developmental abilities. These findings suggest that the creation of specialized interventions, both for parents and children, may fill the possible delays in children’s development probably due to stress, lack of adequate stimulation, or difficult adaptation.

## 1. Introduction

The number of children and adolescents who have survived cancer has increased in recent years due to significant improvements in survival rates [1]. Three main areas crucial in overcoming children’s main developmental tasks were investigated: The cognitive sequelae related to cancer treatments (methotrexate, vincristine, steroids, hematopoietic stem cell transplantation (HSCT) [1,2], motor performance delays [3,4] and the general social impairments related to the illness (academic achievements, interpersonal relationships, and coping skills) [5,6,7].

### 1.1. Cognitive Sequelae Related to the Illness and Its Treatment

Children’s cognitive functioning after being treated with anti-tumoral drugs was largely investigated by analyzing their long-term effects on survivors [8] as deficits in neurocognitive functions may not appear in the immediate period following treatment. Similarly, testing shortly after diagnosis is not feasible [9]. Verbal competence, main executive functions and complex visual-spatial tasks have been found to be impaired in childhood survivors of ALL (acute lymphoblastic leukemia) [10], with a lower performance in measures of working memory than controls [11] and a decline in intelligence and academic achievement [12]. Language performance remained stable in ALL children following intrathecal chemotherapy (ITC) over a two-year period [2], even if this treatment may impact language skills in the long-term [13]. 

The main risk factors in possible developmental deficits in childhood cancer survivors were identified as: Aged under five at diagnosis [14], higher intensity therapy, and the number of years since the individual’s first therapy [15]. Reduced working memory and nonverbal abilities may be present during the first year of treatment after ALL depending on the last methotrexate dose and/or infusion rate [16]. Additionally, attentional dysfunctions are found in survivors of childhood ALL, especially in cases of severe treatment dosages [17]. The cognitive sequelae in children with leukemia can also be influenced by HSCT, with a decline in motor and mnemonic abilities within the first year post-HSCT [1] and in verbal skills, with arithmetic and motor scores attested below the norms by three years post-HSCT [18]. 

### 1.2. Motor Performance Delays

Motor performance has been recognized as a key element for children’s healthy development, especially towards their future social life, and even more so in kindergarten children [19]. Motor competence in early childhood impacts future developmental steps throughout school, stressing an important association between academic and social functioning. Children with cancer showed reduced motor performance at the end of the acute treatment phase, specifically in muscular explosive strength, handgrip strength, leg fatigue, visual-motor coordination, balance, speed, and flexibility [4]. These difficulties appeared to persist in varying degrees several years after the end of treatment [20], in addition to visual–motor deficits and associated difficulties in math and reading achievements [21]. Higher levels of fatigue and lower general wellbeing were self-reported in adult and adolescent (AYA) cancer survivors who underwent HSCT [6]. A known complication of treatment with vincristine (VCR) was the development of polyneuropathy, which can result in the loss of peripheral muscle power in both the upper and lower extremities, with increased motor problems. However, there were significant improvements over time, as revealed by the lower prevalence of neuropathy at increasing intervals following VCR injections [22]. In other studies (i.e., [23]), no correlation was found between motor performance and the cumulative dose of chemotherapy drugs, age, and follow-up time.

Important delays in the motor abilities of preschool children with leukemia during the maintenance phase of therapy were found, especially if children underwent HSCT [24]. These conditions could influence their general social wellbeing and academic achievement, as demonstrated in Section 1.3.

### 1.3. Social Impairments and Academic Difficulties

The experience of illness and its related aspects—such as hospitalization and the overhaul of daily routines—may impact on social functioning as well. Coming back to or beginning schooling after strong medical treatment can be academically and socially difficult for children with cancer. The decline in intelligence and academic achievement appeared to be related to poor social functioning, especially in female children [7]. Peer socialization was reported as the main difficulty by survivors, who displayed limited comprehension of simple social rules (i.e., turn-taking) or with easy reported bounds with older children or teachers than with their peers [25]. Social skills were less developed due to reduced peer interaction [26] and the perceived social support from friends was lower than healthy peers [27], with a reduced ability to maintain friendships and social competence, with survivors demonstrating a more compromised relationship with their best friend [28], in addition to increased self-esteem problems [5].

### 1.4. Aims

Little is known about the developmental trajectories of preschool patients with leukemia, which allow them to have adequate functioning during acute cancer treatments. In this study, we will focus on the specific developmental domains of children with leukemia and compare their adaptive functioning skills with those of a group of healthy peers. By identifying the possible developmental delays in pediatric oncologic patients, we could discover specific indications for what kind of psychological and psychomotor interventions should be implemented to target appropriate remediation. 

The research questions are:Are there differences between the clinical and control groups in their developmental task performance?Are there differences in developmental tasks throughout the different age groups (between ages three and five years)?In which domains do children show more difficulties (communication abilities, daily living skills, socialization competence, motor performance) when compared with Italian norms?In which VABS’ cluster domains and items did children show more difficulties?Are disease and treatment variables associated with children’s developmental delays?

## 2. Materials and Methods

### 2.1. Procedure

Ethical approval was obtained from the University Hospital of Padua Ethical (code 1783P), following the rules of the Declaration of Helsinki of 1975. A clinical psychologist contacted families during the first hospitalization of their children, in the second week after diagnosis. The project aims were explained, and informed consent was obtained. Approximately one year later, the clinical psychologist administered the Vineland adaptive behavior scales (VABS) [29] at the day hospital of the clinic. The intention was to evaluate the adaptive behavior when the children would come back to their social agencies and normal daily routines. After the diagnosis, the data could be influenced by the trauma and the new hospital routine could disrupt their precedent daily routine.

### 2.2. Participants

Participants were preschool children aged 2.5–6 years from an ample sample consisting of 75 children one-year post diagnosis. Of these, 15 exited the study due to death or a terminal diagnosis (N = 9), or otherwise dropped out (N = 6). The response rate attested to 92%, excluding the deceased patients. 

Forty-four healthy peers were matched with the remaining 60 patients for this study, so the study was run on 44 pediatric leukemic patients matched to 44 healthy controls paired for age and gender. Children’s mean age was 4.52 years (SD = 0.94), 19 males and 25 females, all diagnosed with ALL (N = 44), with a mean hospitalization period of 43.85 days (SD = 15.39). All parents were Caucasian, aged 36.97 years on average (SD = 67.10) and had a mean of 12.38 years of schooling (SD = 3.71). Parents’ perceived economic condition was mostly average (51.2%), equally distributed between high (24.4%) and low (24.4%) for Italian norms, but all were above poverty. Families were composed of either two (N = 30), one (N = 11) or three (N = 3) children.

The eligibility criteria for the control group participants (N = 44) was: Absence of life-threatening or chronic illness and no presence of learning or sensory problems and other pathological aspects. The control group of healthy peers consisted of those enrolled at pediatricians’ ambulatories.

The eligibility criteria for the clinical sample were: A diagnosis of acute lymphoblastic leukemia, no relapses or second tumor, speaking Italian language, age cut-offs of <6 years old.

The mean age of the clinical group was 4.52 years (SD = 1.06, range 2.5–6) and the mean age of the control group was 4.5 (SD = 1.06, range 2–5.91). After comparing the two groups based on their mothers’ characteristics, we found that they were homogeneous on their mothers’ age (Z = −0.69, *p* > 0.05) and the number of sons in the family (t47 = −1.60, *p* > 0.05), whilst they differed on their mothers’ schooling years (t47 = −0.16, *p* = 0.002), with mothers of the control group showing higher years of schooling (mean = 15.11, SD = 3.22) than the clinical group (mean = 12.38, SD = 3.72). 

### 2.3. Instruments

The VABS is an interview carried out with parents by a trained psychologist. This interview is psychometrically validated and assesses several adaptive behaviors of children that allow health specialists to have a rich evaluation of developmental pathways, also in children with ALL. The scoring was norm-referenced and referred to specific developmental levels between birth and adulthood along several domains. The 540 items that constituted this interview are organized around four adaptive behavior domains (communication, daily living skills, socialization, and motor skills) and are grouped in clusters. These groupings are distributed in developmental order under sub-domains. The three communication sub-domains are: Receptive, expressive and written language. Personal, domestic, and community compose the daily living skills domain, while the socialization sub-domains are interpersonal, play and leisure, and coping skills. Finally, gross and fine motor abilities are included in the motor abilities domains. Each sub-domain contains a series of items grouped into their representative clusters. The clusters assessed in the clinical and control groups demonstrated a roofing effect and obtained significantly different results in some specific clusters as shown in Table 1.

The items scoring is as follows: “2” stands for behaviors usually or habitually performed, “1” is assigned when behaviors are sometimes/partly performed, when the parent does not know if the child performs the activity or if the child has never had the opportunity to do it, while “0” stands for behavior never performed. The manuals provide users with instructions for scoring caregiver responses. Domain and subdomain raw scores are obtained by summing the numerical values of the responses. The sum of the domain standard scores is used to obtain the composite standard score.

The adaptive behavior scales emphasize the developmental qualities of the construct by providing age-based norms and items that cover a wide range of developmental activities such as: Cognitive, communication and academic skills (i.e., conceptual skills), social competence skills (i.e., social skills), and independent living skills (i.e., practical skills).

Furthermore, medical and socio-demographic information was collected.

### 2.4. Statistical Methods

Descriptive statistics were run to show the child’s developmental skills scores one-year post-diagnosis, also comparing them with age-equivalent norms, specifically, their global score and the scores related to each VABS subscale.

The possible socio-demographic differences between the two samples (clinical and control) were investigated by adopting paired-samples Wilcoxon tests to estimate the comparability of the samples. Inferential comparisons between cancer patients and control samples, matched by gender and age, were run by adopting paired-samples Wilcoxon tests. Spearman’s correlations were run to understand if medical variables could impact on VABS domains and subdomains. The evaluated statistical significance was attested at the nominal *p* < 0.05 level. All data were analyzed using SPSS Version 20 (SPSS Inc., Chicago, IL, USA).

## 3. Results

### 3.1. Developmental Domains in Children with Leukaemia Compared with Healthy Peers

Parents of children with leukemia reported significantly lower developmental scores in their children compared to healthy peers, both in composite (Z= −5.31, *p* = 0.0001) and in two out of the four adaptive domains: Communication (Z= −4.46, *p* = 0.0001), and socialization (Z= −3.80, *p* = 0.0001). For communication, significant differences were identified in the following sub-domains: Receptive (Z = −4.49, *p* = 0.0001) and expression (t47 = −4.81, *p* = 0.0001). Socialization sub-domains were all significantly lower in children with leukemia: Interpersonal relationships (Z = −3.13, *p* = 0.0001), and coping skills (Z = −2.88, *p* = 0.004). (Table 2 and Figure 1).

We also have to take into consideration that the mothers’ education was different when comparing the clinical and control groups, and that this variable could be associated with the children’s developmental tasks. We, therefore, ran a series of Spearman’s correlations in the clinical and control groups between mothers’ schooling years and VABS domain scores. None of these statistical analyses obtained significance (*p* > 0.05). We concluded that this variable does not have a significant association with children’s developmental tasks (Table 3).

### 3.2. Differences in Developmental Skill Performance in Age Groups (30–48 Months, 49–60 Months, 61–71 Months)

We divided the children into three age groups: 30–48 months (N = 13), 49–60 months (N = 17) and 61–71 months (N = 14) to better investigate developmental skills by age. We ran the paired sample Wilcoxon-test in each age band. A significant difference between the clinical and control groups was obtained for the VABS composite score of children aged 30–48 months (Z = −2.98, *p* = 0.003), of those aged 49–60 months (Z = −3.24, *p* = 0.001) and of those aged 61–71 months (Z = −2.97, *p* = 0.03) with the clinical group showing lower scores than the control one. 

Reduced communication abilities were recognized in the clinical group of children aged 30–48 months (Z = −2.44, *p* = 0.01), in particular in the receptive (Z = −2.76, *p* = 0.006) and expressive (Z = −2.48, *p* = 0.01) subscales. The same result was obtained in those aged 49–60 months (Z = −2.79, *p* = 0.005), specifically in the receptive (Z = −3.06, *p* = 0.002) and expressive (Z = −3.24, *p* = 0.001) subscales. Moreover, children aged 61–71 months belonging to the clinical group obtained lower scores in communication abilities (Z = −2.94, *p* = 0.03), especially in the expressive (Z = −2.85, *p* = 0.04) and in the written scales (Z = −1.9, *p* = 0.05).

Parents of children aged 30–48 months belonging to the clinical group reported significantly lower Interpersonal relationships (Z = −3.06, *p* = 0.002). Moreover, the group of children aged 49–60 months showed lower scores in the play and leisure time subscale (Z = −2.40, *p* = 0.01). Figure 2 shows these results.

### 3.3. In Which Cluster Belonging to Communication and to Socialization Domains Did Children Show More Difficulties?

Children with leukemia showed significantly lower levels of development, according to parental perceptions, with respect to some items grouped into specific clusters. We ran the Wilcoxon paired t-tests to evaluate the statistical differences between the clinical and control groups in each cluster. Table 4 documents only significant results.

From Table 4, we can see how preschool patients, when compared with healthy peers, showed difficulties exclusively in the attentive functions involved in the receptive sub-domain. When dealing with expressive skills, we found several significant difficulties: Articulation and recitation, use of plurals and verbs in different tenses and children’s capacity to provide information about himself/herself. 

Regarding the socialization domain, we analyzed the possible differences in the Interpersonal relationships cluster. Recognition of emotion seemed to be unaffected, even if the pediatric patient failed to recognize or verbally classify his/her own joy, sadness, fear or anger. The identification of people and the first forms of social communication seemed to be problematic, while giving presents and maintaining friendships is largely unaffected, except for the giving of little presents to family members, the preference for some friends and the presence of a darling friend.

The Play and Leisure subscale showed lower clusters in the clinical sample, especially when following/respecting the rules of play, participating in games and going out with friends. Sharing/cooperation and watching TV were relatively unaffected, except sharing toys or other personal items with or without being reminded to do so and the recognition of the name of at least one favorite TV program and the day and channel on which it is broadcasted. Furthermore, the Coping skills subscale presented lower levels, specifically about respect for the rules, conversational rules, and responsible time management.

### 3.4. Pre-School Patients’ Delays Compared with Norms

We scored delays of almost three months of difference between T scores and chronologic age for each domain and subdomains of patients’ group. Figure 3 shows these results. Considering the VABS domains socialization and motor abilities seemed to be more affected in their normal functioning. Regarding communication, the mostly affected subscales were written and in the daily living skills the community rules. 

### 3.5. Disease and Treatment Variables Associated with Children’s Adaptive Functioning

We run a Mann−Whitney test to understand if the risk band (standard and medium risk) influence the VABS domains and subdomains. We did not obtain any significant difference. We run also Spearman’s correlations between days of hospitalizations and therapy’s sum of toxicities and the VABS scores. Only a significant correlation was found between communication score and toxicities’ sum (r = 0.32, *p* = 0.05), especially the written scale (r = 0.46, *p* = 0.006).

## 4. Discussion

Early childhood is a crucial time of life, where basic adaptive skills experience rapid and dynamic growth, significantly impacting social and academic learning. The child has to overcome different daily tasks, such as “feeding oneself, maintaining hygiene (by washing hands, brushing teeth, and bathing), changing a variety of clothes, and controlling bowel and bladder” [30]. The child also has to develop gross motor abilities, including to gamble, catching/throwing the ball and riding a tricycle or bicycle. Furthermore, important fine motor skills should be developed, such as gathering objects, making models, drawing and opening drawers or doors. Language activities during preschool years include attention tasks, vocabulary level, use of language in sentences, use of proper names/plurals and verb tenses, question formulation, the use of abstract concepts and the type of articulation. Children at this stage learn to recount their own experiences and provide information about themselves, but also to take turns and negotiate social interactions. However, these conceptual, social and practical skills could be negatively influenced by cancer experiences and treatments, so much so that delays in their psycho-social and motor competencies may occur.

Negative treatment sequelae in children with leukemia can be attributable to different drugs [31]: i.e., mucositis is due to daunoblastina, where possible neuropathy could be caused by VCR. Steroids could lead to humor and behavioral difficulties, psychosis, bone fragility, myopathies, eye problems and neuropathies, while Peg-asparaginase could cause trembling fingers and gastrointestinal disorders. Nausea and vomit could be attributable to ifosfamide, while working memory and attention difficulties could be caused by intrathecal methotrexate. These negative treatment effects could contribute to increased developmental delays, together with other aforementioned hospitalization stressors.

Research on the development of children with cancer has focused less on the overcoming process of developmental tasks in early childhood during cancer treatments and instead remains in the off- therapy phase.

In this study, preschool children receiving therapy for leukemia showed important developmental difficulties and delays that could contribute to maladaptive personal growth. The major limitations of these children, compared with matched healthy peers, were found in communication, socialization and motor skills and in the global adaptive functioning throughout the several age groups. Children aged 30–48 months and those belonging to 49–60 months’ group showed significant difficulties in communication abilities, especially receptive and expressive scores, and in socialization skills, especially the interpersonal relationships. In the 61–71 months age range, the main limitations were identified in the written and expressive skills regarding communication and play and leisure for socialization domain.

At this purpose, it becomes fundamental to identify children more at risk for adaptive skills delays. Previous studies [24,32] identified important delays in children in their gross and fine motor performance just during the maintenance therapy phase and also after completion of therapies, with the frequency of days of hospitalization and HSCT experience that could drastically dampen all the children’s adaptive skills.

This study also facilitates a focus on the main difficulties of children, as reported by parents, adopting a qualitative approach, allowing detailed information to be collected on the specific clusters of each domain and sub-domain.

Attention functions in receptive domains were reduced, like, for example, the capacity of listening to the teacher/a story/lesson for a certain period of time, ranging from five to 30 min. This study tries to understand if the disease and the treatment could reduce executive functions [17], with possible consequent difficulties in academic achievement when children start school and positive self-esteem perceptions [33]. We checked these treatments (chemotherapy toxicities, days of hospitalization) and diseases variables (risk band category) in our analyses that showed no differences in developmental domains along these. Probably, it is more the stress and the difficult adaptation associated with this condition that could contribute to impairments in the development, not strictly chemotherapy or CNS treatments.

Confirming the literature on childhood cancer survivors on reduced verbal competence and processing speed and attention in ALL [13], parents reported significantly reduced expressive functions. Language form is less developed, such as the use of connectors, plural and verb tenses could be incorrect and the articulation of words could be inaccurate, with letters or sounds being confounded. 

In the medical setting, children very often became silent both with the pediatric staff and even with parents. They either could not speak due to trauma or because they feared to ask questions about their illness. The adults could also not correct their children’s language mistakes because their parenting attitude was more comprehensive for children’s difficult health conditions. Children with leukemia were very often isolated from their peers (e.g., kindergarten activities and park visits), and in the hospital, it was difficult to play with other patients as children in other settings would. The medical setting and hospitalization could potentially cause such language delays.

On the other hand, the capacity to recount their own experiences and provide information about themselves, such as talking about their experiences using a detailed narrative form, was intact, together with the ability to express complex ideas. Their receptive and expressive abilities seemed to be unaffected, while correct formulation and articulation did not always escape unharmed. 

Studies involving social behavior and peer relationships generally concluded that children with leukemia were more sensitive and isolated than peers, thus developing social competence limitations [26]. In this study, children with leukemia confirmed socialization difficulties compared with healthy peers. Specifically, they had limitations in their interpersonal relationships and in the play and leisure activities, particularly in younger children aged 30–60 months: Even if their recognition of emotions was maintained, they did not recognize or verbally classify their own feelings, had difficulties identifying people and expressing initial social communications, with a consequent impediment to make a preference for some friends or for a darling friend. Coping skills attested at lower levels, specifically respecting the rules, conversational turns, and responsible time management.

Impairments in motor adaptive skills were identified in children with leukemia during the therapy [24]. This study confirmed difficulties in gross and fine motor abilities comparing VABS equivalent scores with aged-norms. Generally, there is some impact of therapies on motor abilities, but it is principally on gross motor skills. In the clinic, during hospitalization, there is the possibility of playing with the volunteers (wooden buildings, work with play dough or clay, and decoupage) and parents are helped by psychologists to stimulate their children at home (i.e., to cook, hang out the laundry, draw or perform daily living activities). However, being persistently bedridden and the associated fatigue could impair gross motor achievements, subsequently undermining muscle strength and balance, both during the therapies [20] and after HSCT [33]. Fine motor skills may display a long-term delay, for example, after therapy or HSCT, as documented by Taverna and colleagues [32]. Moreover, written skills are underdeveloped, especially in the older children aged 61–71 months, just in the last kindergarten year before school, when this ability should be reached as part of expected school readiness skills. 

One limit of this study is that children are not very numerous, making difficult to generalize the results of the present work, even if they are all children with ALL with a standard and a medium risk of treatment. Due to the small number of participants, we decided to adopt non-parametric tests for statistical analyses.

We have not any baseline measure of children’s adaptive skills before or close to the cancer diagnosis. However, it would have been impossible to assess the adaptive behavior prior to the illness, and, similarly, it would have been very difficult to have the parents’ collaboration and their valid reports immediately after the diagnosis, when the therapies begin, as it is a very critical time. After the diagnosis, the data could be staggered by the trauma and the new hospital routine could disrupt their precedent daily routine. The intention is to evaluate the adaptive behavior when the children could come back to their social agencies and normal daily routines. Since the children were evaluated only one-year post-diagnosis, it would also have been of great interest to compare changes in the areas tested from baseline both in the patients with leukemia and the controls, instead of this limited cross-sectional approach. Future studies should explore these domains with a longer follow-up of pre-school children.

One strength of this study is the in-depth semi-structured interviews with parents of preschool patients during the maintenance phase of therapy, when the children can partially re-enter their normal daily routines, meeting peers after isolation and beginning primary school. This is the first study that focused specifically on adaptive skills in childhood leukemia patients of preschool age during therapies. Moreover, the comparison with a control group of healthy peers has allowed to identify the degree of delays, helped by the innovative use of qualitative information derived from the VABS clusters, which facilitate the understanding of specific developmental difficulties to create psycho-educative interventions.

## 5. Conclusions

Based on these results, the following clinical suggestions are proposed.

Firstly, we have to take into consideration children’s age for the possible psychological interventions, because we have seen, after reviewing the literature and partially in our study, how age influenced both the child’s ability to cope and adaptability and, consequently, their quality of life. The 30–48 months-old children seemed to be more at risk for receptive and interpersonal relationships’ developmental delays, while 49–60 months-old children had more delays in expressive and play and leisure activities, and the last group of older children aged 61–71 months had more written and verbal delays. The intervention could be set up ad hoc for each age group with specifically empowered stimulation on language, or on written or on socialization domains. For example, particular attention could be devoted to those aspects of communication, receptive, and expressive, which will allow the child to interact with peers once they return to normal life. Linguistic stimulation should certainly be oriented to lexical and syntactic enrichment, without however losing sight of the need to use communication to express one’s emotions, fears and difficulties related to the contingent situation. Specific stimulation exercises should, therefore, be conducted not only in a didactic perspective of maintaining or implementing skills, but rather according to a functional perspective in which language is primarily a vehicle of communication and social interaction. Looking at the healthy part of the child and his ability to return to everyday life and school learning also requires preparing interventions aimed at stimulating those skills that in normal contexts are developed in kindergarten. Thinking about the future of children suffering from leukemia means taking care to put them in a position to face the new learning challenges with sufficient school readiness and fine motor skills. For this purpose, it would be advisable to introduce simple psycho-motor activities, suitable for the hospital context, which allow the child to develop an adequate grip on the graphic tools, coordinate the vision with the graphic execution, learn to write his/her name, and know some alphabet letters.

Secondly, our empirical results can help to set up specialized interventions focused on parents and children to meet the developmental difficulties associated with leukemia. In particular, receptive attention problems, reduced language in sentences and in questions together with interpersonal relationship difficulties, social, and play rules and fine and gross motor problems were the main compromised developmental domains. Consequently, specific language and psycho-motor programs can be implemented during hospitalization. Socialization and educational programs can be proposed, both during the acute phase of treatment and day-hospital follow-ups. Social plays and educative guidelines can be taught to parents to stimulate their child at home, facilitating their children’s re-entry into their normal routines as soon as possible (school, sport and hobbies). Parents should have also support in their parenting role with the children, considering that they are highly distressed and could develop also post-traumatic stress symptoms [34,35]. 

These proposed approaches to intervention need to be tested and validated throughout the future studies adopting an effective design with three groups of children under these stimulations in three different time points: During the first year of therapies, after the first year, and when they are off therapy. In this way, the empirical data could suggest us the best timing of intervention or also the necessity to have more stimulation activity time. 

## Figures and Tables

**Figure 1 children-06-00092-f001:**
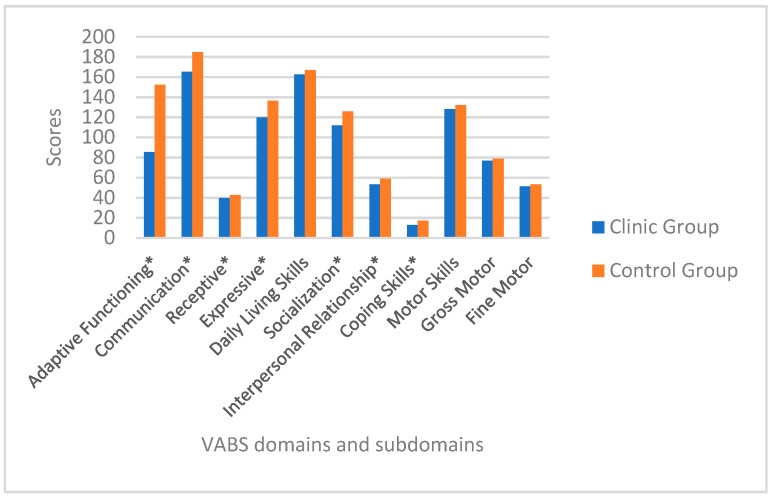
VABS adaptive functioning domain scores comparing: (**a**) Children with leukemia one-year post-diagnosis and controls, (**b**) VABS adaptive functioning sub-domain scores comparing children with leukemia one-year post-diagnosis and healthy peers. * = statistically significant domain and subdomains.

**Figure 2 children-06-00092-f002:**
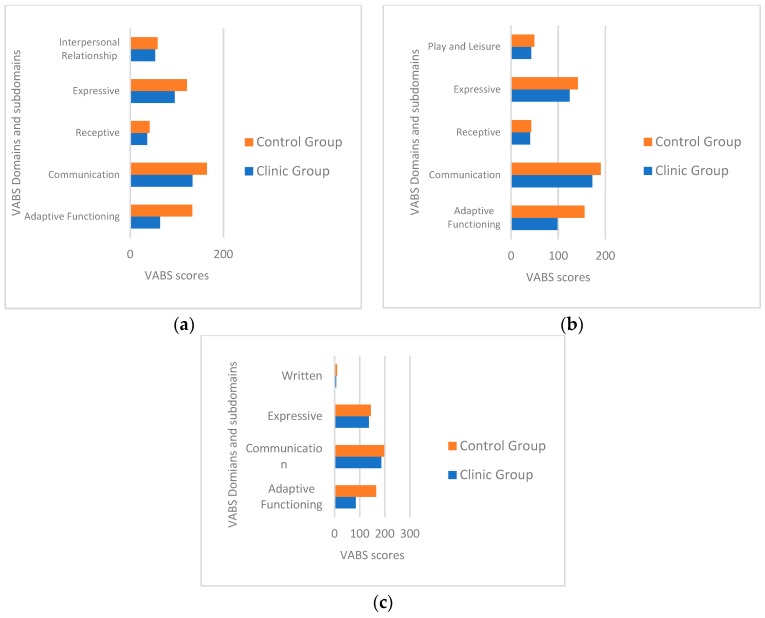
VABS adaptive significative functioning domain and subdomain scores comparing: (**a**) Children with leukemia aged 30–48 months one-year post-treatment and matched controls, (**b**) children with leukemia aged 49–60 months one-year post-treatment and matched controls, (**c**) children with leukemia aged 61–71 months one-year post-treatment and matched controls.

**Figure 3 children-06-00092-f003:**
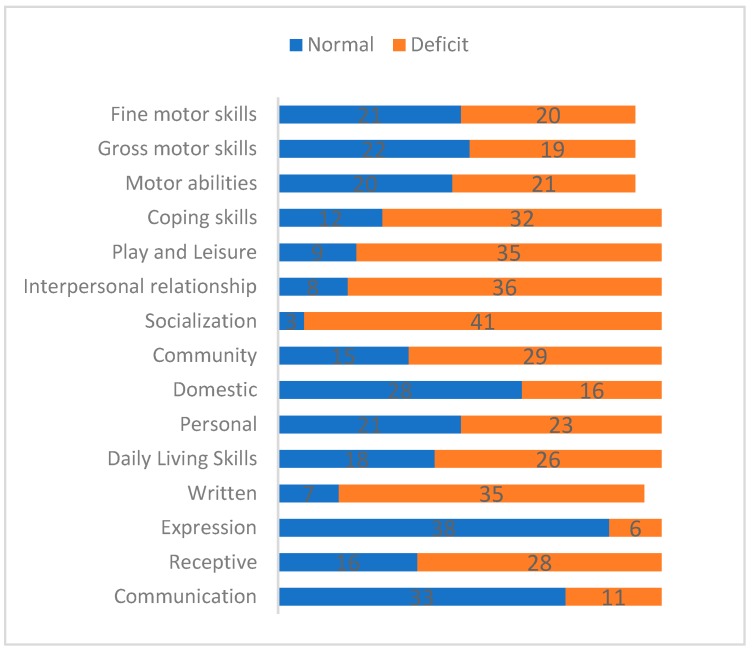
VABS adaptive functioning domain T scores of patients comparing with Italian norms: Deficit and normal categories.

**Table 1 children-06-00092-t001:** Vineland adaptive behavior scales organized by domains, subdomains, clusters and item content: Significant items.

Domain	Subdomain	Cluster	Items content
Communication	Receptive	Attention	How and how long the individual listens to someone, pays attention to activities or stories and understands given information. (E1–E5)
	Expressive	Articulation	Quality and precision in speech articulation are evaluated. (M1–M4)
		Recitation	Ability to recite rhymes, songs, folk tales is assessed. (N1–N4)
		Use of plurals and verbs times	Individual uses verb forms correctly in different tenses. (O1–O6)
		Provide information about yourself	Ability to answer correctly to questions referred to address, telephone number or other personal information is measured. (P1–P7)
Socialization	Interpersonal Relationships	People identification	Individual knows names of family members and identifies them through features other than their name. (F1–F4)
		First forms of social communication	Individual is able to participate in conversation. (G1–G3)
		Friendships	Individual has friendships of the same of other sex. (H1–H5)
	Play and Leisure	Games participation	Participation in different types of games (i.e., cards, hazard-based board) is assessed. (G1–G4)
		Go out with friends	Individual is able to meet friends outside home in the afternoon or evening. (H1–H4)
	Coping skills	Respect for the rules	Individual respects rules in community and social situations. (A1–A4)
		Good education in conversation	Individual is able to have conversations with others showing good education. (C1–C3)
		Responsible time management	Individual shows ability in managing time limits and making projects along time. (D1–D2)

**Table 2 children-06-00092-t002:** Adaptive functioning in children with leukemia and healthy controls: Mean and standard deviations.

Scales	Leukaemic Children		Healthy Peers	
	Mean	SD	Mean	SD
Adaptive functioning composite	85.53	51.53	152.34	20.16
Communication	165.34	184.77	184.77	20.58
Receptive	39.56	4.16	42.72	1.70
Expressive	119.90	26.75	136.50	16.56
Socialization	111.93	27.68	125.72	24.79
Interpersonal Relationship	53.40	10.72	58.75	6.87
Coping Skills	12.97	8.23	17.20	9.55

**Table 3 children-06-00092-t003:** Spearman’s correlations between maternal education and developmental skills.

	Mother’s Schooling Years
VABS composite	r = −0.046
	*p* = 0.77
VABS Communication	r = −0.030
	*p* = 0.84
VABS Socialization	r = 0.19
	*p* = 0.31
VABS Motor abilities	r = 0.027
	*p* = 0.86

**Table 4 children-06-00092-t004:** Clinic and control group comparisons along VABS items, grouped by cluster belonging to communication and socialization domains (ns stays for not significant).

		Patients		Healthy Peers		Statistical Analyses	
Sub-domain, Cluster	Item	M	SD	M	SD	Z	*p*
Receptive, Attention	E1	1.4	0.81	1.95	0.21	−4.54	0.001
	E2	1.22	0.93	1.56	0.66	−2.15	0.003
	E3	0.9	0.91	1.68	0.51	−6.20	0.0001
	E4	0.79	0.87	1.63	0.68	−5.85	0.0001
	E5	0.15	0.52	0.04	0.3	1.22	0.23 ns
Expressive, Articulation	M1	1.54	0.76	1.9	0.42	−2.55	0.01
	M2	1.54	0.76	1.88	0.44	−2.44	0.01
	M3	1.38	0.86	1.9	0.42	−3.1	0.002
	M4	1.27	0.89	1.59	0.81	−1.79	0.07 ns
Expressive, Recitation	N1	1.68	0.73	1.86	0.51	−1.41	0.15 ns
	N2	1.56	0.81	1.88	0.44	−2.32	0.02
	N3	0.75	0.94	1.84	0.52	−4.65	0.0001
	N4	1.09	0.98	1.68	0.67	−3.1	0.01
Expressive, Use of plurals and verbs times	O1	1.2	0.95	1.88	0.44	−3.83	0.0001
	O2	1.15	0.96	1.81	0.58	−3.48	0.0001
	O3	0.97	0.97	1.75	0.57	−4.10	0.0001
	O4	0.86	0.97	1.72	0.69	−4.30	0.0001
	O5	0.27	0.62	1.06	0.79	−4.44	0.0001
	O6	0.36	0.71	1.45	0.79	−4.87	0.0001
Expressive, Provide information about yourself	P1	1.45	0.9	1.9	0.42	−3.16	0.002
	P2	1.45	0.9	1.81	0.54	−2.81	0.005
	P3	1.31	0.95	1.88	0.44	−3.50	0.0001
	P4	0.93	0.99	1.38	0.92	−2.67	0.008
	P5	0.43	0.81	0.7	0.87	−1.34	0.18 ns
	P6	0.13	0.5	0.18	0.58	−0.45	0.65 ns
	P7	0.36	0.78	1.22	0.86	−4.08	0.0001
Interpersonal relationships, People identification	F1	1.68	0.73	2	0	−2.64	0.008
	F2	1.63	0.78	2	0	−2.82	0.005
	F3	1.43	0.87	1.86	0.44	−2.73	0.006
	F4	1.31	0.93	1.88	0.44	−3.50	0.0001
Interpersonal relationships, First forms of social communication	G1	1.22	0.93	1.65	0.71	−3.04	0.002
	G2	1.27	0.97	1.61	0.78	−2.10	0.035
	G3	0.81	0.97	0.81	0.97	0.15	0.88 ns
Interpersonal relationships, Friendships	H1	1.04	1.01	1.5	0.87	−2.35	0.018
	H2	1	1.01	1.25	0.26	−1.39	0.16 ns
	H3	0.72	0.97	1.45	0.9	−3.41	0.001
	H4	0.06	0.33	0.22	0.64	−1.53	0.12 ns
	H5	0.59	0.92	0.86	1	−1.50	0.13 ns
Play and Leisure, Following play rules	G1	0.18	0.58	1.52	0.82	−2.28	0.022
	G2	0.9	1	1.38	0.89	−2.93	0.003
	G3	0.75	0.96	1.04	0.96	−2.03	0.042
	G4	0.7	0.95	1.13	0.92	−2.32	0.021
Play and Leisure, Games participation	H1	0.43	0.81	0.97	0.99	−3.20	0.001
	H2	0.4	0.81	0.86	1	−2.89	0.004
	H3	0.18	0.58	0.43	0.81	−1.96	0.05
	H4	0.13	0.5	0.5	0.87	−2.53	0.011
Play and Leisure, Go out with friends	I1	0	0	0.27	0.69	−2.50	0.014
	I2	0	0	0.02	0.15	−1	0.32 ns
	I3	0	0	0.18	0.58	−2	0.046
Coping skills, Respect for the rules	A1	1.56	0.62	1.84	0.37	−2.35	0.02
	A2	1.36	0.68	1.93	0.25	−4.18	0.0001
	A3	1.65	0.6	1.7	0.63	−0.19	0.84 ns
	A4	1.43	0.78	1.93	0.25	−3.38	0.001
Coping skills, Good education in conversation	C1	0.25	0.61	0.75	0.91	−3.03	0.002
	C2	0.13	0.4	0.5	0.76	−2.62	0.009
	C3	0.51	0.52	0.5	0.87	−2.15	0.03
Coping skills, Responsible time management	D1	0	0	0.41	0.81	−3	0.003
	D2	0	0	0.25	0.65	−2.33	0.02

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
