# Peer review of "The Developmental Pathways of Preschool Children with Acute Lymphoblastic Leukemia: Communicative and Social Sequelae One Year after Treatment"

_children, 2019, doi:10.3390/children6080092_

Round 1
Reviewer 1 Report
Figure 1 would be more clear with the subdomans under each broader domain (rather than in a separate figure (b)). The provision of p-values (or an astericks) denoting those that were statistically significant would also improve the quality of this figure.
The first table 3 (should be labeled table 2): Spearman’s correlations between socio-demographic variables and developmental skills should be titled maternal education as it does not appear that you compared any other socio-demographic variables. This can be provided in a supplement.
The second table 3 should include the some specification of what the Italian norms means are for comparison purposes. In addition, the title of the second column should be titled Health Peers. I do not think it is necessary to provide the score for each item since the details of the VABS items are not provided. Can a single score be provided for each category?
The conclusion provides more detailed examples of possible interventions. Additional details about whether these proposed approaches to intervention need to be tested and validated or whether they are already well accepted/evidence based interventions would be helpful. Likewise, the timing of these interventions (prior to one year, after one year, or once patient is off thereapy) would also be helpful.
Author Response
Comments and Suggestions for Authors
Figure 1 would be more clear with the subdomains under each broader domain (rather than in a separate figure (b)). The provision of p-values (or an astericks) denoting those that were statistically significant would also improve the quality of this figure.Reply 1: Figure 1 was modified along your suggestions
The first table 3 (should be labelled table 2): Spearman’s correlations between socio-demographic variables and developmental skills should be titled maternal education as it does not appear that you compared any other socio-demographic variables. This can be provided in a supplement.
Reply 2: we changed the title of Table 3 as follows: “Spearman’s correlations between maternal education and developmental skills” and we adjusted the labels of the several tables throughout the text.
The second table 3 should include the some specification of what the Italian norms means are for comparison purposes. In addition, the title of the second column should be titled Health Peers. I do not think it is necessary to provide the score for each item since the details of the VABS items are not provided. Can a single score be provided for each category?Reply 3: The second table 3 (actually table 4) showed the comparing analysis between patients and healthy peers along the several clusters (group of item). In this case we don’t show the normative data, that are placed in the next paragraph (3.4). We changed the title of the second column. We preferred to give the score for each item belonging to the cluster, because this instrument doesn’t show a global cluster score. However, we reduced the table inserting only the items with significance so to make the table readable.
The conclusion provides more detailed examples of possible interventions. Additional details about whether these proposed approaches to intervention need to be tested and validated or whether they are already well accepted/evidence based interventions would be helpful. Likewise, the timing of these interventions (prior to one year, after one year, or once patient is off therapy) would also be helpful.Reply 3: We added this information. See lines 466-470 at page 14
Reviewer 2 Report
The authors comprehensively evaluated the developmental delays in preschool children with acute lymphoblastic leukemia (ALL) one year after treatment by using the Vineland Adaptive Behavior Scales (VABS). They showed differences in the developmental delays by the age groups. These findings indicate appropriate intervention should be made according to the patient's age. Although the number of enrolled patients and parents were not so many and the studying period was short, this manuscript show important findings.
Major points
In the legends to Figure 2 and line 324, the age groups are described as 30-48, 49-60, and 61-71 months: however, in lines 410 and 411, these are described as 30-41 and 42-60 months. The authors should correct the discrepancy.
The authors should explain how the VABS is universally and popularly accepted so that the reader who are not always familiar with the evaluation of developmental pathways in children with ALL can readily understand this field.
Minor point
English explanation should be more polished.
Author Response
Comments and Suggestions for Authors
The authors comprehensively evaluated the developmental delays in preschool children with acute lymphoblastic leukemia (ALL) one year after treatment by using the Vineland Adaptive Behavior Scales (VABS). They showed differences in the developmental delays by the age groups. These findings indicate appropriate intervention should be made according to the patient's age. Although the number of enrolled patients and parents were not so many and the studying period was short, this manuscript show important findings.
Thank you
Major points
In the legends to Figure 2 and line 324, the age groups are described as 30-48, 49-60, and 61-71 months: however, in lines 410 and 411, these are described as 30-41 and 42-60 months. The authors should correct the discrepancy.
Thank you. We checked and corrected the age groups. See lines 432-433
The authors should explain how the VABS is universally and popularly accepted so that the reader who are not always familiar with the evaluation of developmental pathways in children with ALL can readily understand this field.
We added a consideration on VABS at lines 149-150 page 4.
Minor point
English explanation should be more polished.
We used a referenced proof-reading English service in the second revision. However, we checked and corrected the English explanation.
This manuscript is a resubmission of an earlier submission. The following is a list of the peer review reports and author responses from that submission.
Round 1
Reviewer 1 Report
In their paper Tremolada, Taverna at alt. presented a study on the effect of cancer therapies in children with leukemia and they try to address the effects of these therapies and some areas of children development compared to healthy peers. They claim that "motor, cognitive and social difficulties related to cancer therapies heavily impact the development of children with cancer". The paper tries to provide insights on these developmental impairments assessed via the Vineland Adaptive Behaviour scale and by performing some paired sample T-test, however the study presents with some major flows
The conclusion that cancer therapies are primarily responsible for these results are highly speculative. First the authors have not performed any experiment themselves to corroborate this theory but they only refer to published scientific literature. However, even in this case their conclusion and interpretation are not always correct. As an examples they say in their discussion that “Attention functions in receptive domains were reduced, like, for example, the capacity of listening to the teacher/a story/lesson for a certain period of time, ranging from five to 30 minutes. This problem could be associated with chemotherapy and CNS treatment, which reduces executive functions [(17)], with possible consequent difficulties in academic achievement when children start school [(33)]”, However, in their paper Buizer AI et alt, (17) claim that the effects on CNS are depended on type of cancer, gender but most importantly dosage of chemotherapy. There is no scientific evidence, or experimental work in the paper from Tremolada, Taverna to support such conclusion.
The authors are not considering in their assessment how the disease itself is affecting the life and development of children. The only scientific way to asses this will be to have a cohort of patients that are not treated (placebo) but clearly this is not feasible. However, this should help the authors to report in their discussion/conclusion only on the results of study they have performed and to avoid speculation on the specific effect of the treatment. It will be probably more appropriate to say that the disease, the treatment and the stress associated with this condition could contribute to impairments in the development, but since they have not performed any biochemical, molecular or genetic test they can’t reach to the conclusion that these defects are only due to cancer treatment.
These observations are strengthen by the fact that the statistical power of the methods applied in this paper is not determined. In many cases, the SD of the case reported are very high making a difference between controls and disease difficult to assess. The authors claims these are significant based on p-value but this is an over statement considering the coefficient of variation between individuals in the population is not provided. Moreover, the cohort considered is very small (48) making these results doubtable. The authors should perform a more accurate analysis and also they will need to provide a file (supplementary) with data from all the cohort analysed to allow readers to gain information from individual cases. Also, they need to provide better details on the statistical approaches (e.g they mention T-test, was this done on excel, R?). The same applies for all other method performed.
In many of their figures there are no unit to clarify what the axis’s are (eg. Figure 1a/b no unit for y axis) or in table 1 they should define in the legend the meaning of SD (e.g standard deviation), T, etc.
Finally, the reference sections should be revisited to make sure things are consistent in English (in many cases the titles of the paper is in English but months of publication or comments are in Italian)
e.g: 1) Shah AJ, Epport K, Azen C, Killen R, Wilson K, De Clerck D, et al. Progressive Declines in
Neurocognitive Function Among Survivors of Hematopoietic Stem Cell Transplantation for PediatricHematologic Malignancies: J Pediatr Hematol Oncol. giugno 2008;30(6):411–8.
.
Reviewer 2 Report
The authors used a cross-sectional design to assess development in preschool-aged children with acute lymphoblastic leukemia one year post-diagnosis in comparison to a control group of healthy children using the validated VABS.
Does the introduction provide sufficient background and include all relevant references?
Introduction provided adequate background to the topic and provided good overview of the pertinent literature.
Is the research design appropriate?
The authors employed a paired t-test, comparing children with leukemia to matched healthy controls. Given that the VABS is validated and norm-referenced, it would have been preferable to additionally compare the scores of the patients with the reference standard for age. The introduction of the control group of healthy children may actually introduce bias, as they are matched only by age and gender, and could have differed enough by other key features to have affected the results (in particular maternal education). Often, healthy siblings are used a match in this setting (though then their ages would differ).
It is somewhat perplexing that patients were identified and consented 1 year prior to VABS evaluation. Given that they are being consented, it is not clear why the VABS could not also be administered at that time point. This would have been a more optimal study design as it would have allowed for within subject comparison (pre- and post- treatment).
Are the methods adequately described?
The manuscript lacked key details about methodologies- including how the VABS was administered and scored, which is crucial for interpreting the results.
Given the substantial differences in therapy between ALL and AML—and likely different psychosocial sequelae—the AML patients should have either been excluded or reported separately (though the small numbers make the latter approach difficult.)
The authors did not explicitly state inclusion criteria, including age cut-offs.
While the authors state that they adjust for multiple comparison, they do not explain the methodology used to adjust. (And it does not appear to be reflected in the results, as they use a p value of 0.05.)
Are the results clearly presented?
Table 2- Please spell out the titles for each column (and explain what d is).
Table 3 was too long to be able to easily discern the relevant information; this could be summarized for ease of understanding and the complete table should be presented as a supplemental table.
The results of the Pearson’s correlation (in relation to maternal SES) should have been included, even if the p value was not statistically significant, as the test could have been underpowered to detect statistical significance, but the results could nonetheless be meaningful.
Are the conclusions supported by the results?
The authors did not adequately explain the significance of their findings, and thus, it is not clear what their research adds to the existing literature.
Rather than summarizing each statistically significant result in the discussion (which read like a duplication of the results), it would have been preferred if they had instead interpreted these results, and presented plausible mechanisms for the differences that were seen. This might be improved by discussing prominent findings in domains/ sub-domains, rather than each item of the VABS.
It is not clear if the differences that they describe are significant and/ or clinically meaningful. An emphasis on the differences with the greatest magnitude may help direct future attention or interventions to specific developmental areas.